# Molecular Plasmonic Silver Forests for the Photocatalytic-Driven Sensing Platforms

**DOI:** 10.3390/nano13050923

**Published:** 2023-03-02

**Authors:** Maxim Fatkullin, Raul D. Rodriguez, Ilia Petrov, Nelson E. Villa, Anna Lipovka, Maria Gridina, Gennadiy Murastov, Anna Chernova, Evgenii Plotnikov, Andrey Averkiev, Dmitry Cheshev, Oleg Semyonov, Fedor Gubarev, Konstantin Brazovskiy, Wenbo Sheng, Ihsan Amin, Jianxi Liu, Xin Jia, Evgeniya Sheremet

**Affiliations:** 1Research School of Chemistry & Applied Biomedical Sciences, Tomsk Polytechnic University, 30 Lenin Ave, 634050 Tomsk, Russia; 2Montanuniversität Leoben, Franz Josef-Straße 18, 8700 Leoben, Austria; 3State Key Laboratory of Solid Lubrication, Lanzhou Institute of Chemical Physics, Chinese Academy of Sciences, Lanzhou 730000, China; 4Van’t Hoff Institute of Molecular Science, University of Amsterdam, Science Park 904, 1098 XH Amsterdam, The Netherlands; 5School of Materials Science and Engineering, Northwestern Polytechnical University, Xi’an 710072, China; 6School of Chemistry and Chemical Engineering, Shihezi University, Shihezi 832003, China

**Keywords:** laser fabrication, chemical sensors, flexible electronics, SERS, photocatalysis, 4-nitrobenzenethiol, cancer

## Abstract

Structural electronics, as well as flexible and wearable devices are applications that are possible by merging polymers with metal nanoparticles. However, using conventional technologies, it is challenging to fabricate plasmonic structures that remain flexible. We developed three-dimensional (3D) plasmonic nanostructures/polymer sensors via single-step laser processing and further functionalization with 4-nitrobenzenethiol (4-NBT) as a molecular probe. These sensors allow ultrasensitive detection with surface-enhanced Raman spectroscopy (SERS). We tracked the 4-NBT plasmonic enhancement and changes in its vibrational spectrum under the chemical environment perturbations. As a model system, we investigated the sensor’s performance when exposed to prostate cancer cells’ media over 7 days showing the possibility of identifying the cell death reflected in the environment through the effects on the 4-NBT probe. Thus, the fabricated sensor could have an impact on the monitoring of the cancer treatment process. Moreover, the laser-driven nanoparticles/polymer intermixing resulted in a free-form electrically conductive composite that withstands over 1000 bending cycles without losing electrical properties. Our results bridge the gap between plasmonic sensing with SERS and flexible electronics in a scalable, energy-efficient, inexpensive, and environmentally friendly way.

## 1. Introduction

Plasmonic nanostructures as the substrates for surface-enhanced Raman spectroscopy (SERS) sensing of different analytes have been a hot topic for decades [1,2,3,4], and lately, they have been intensively investigated in the context of biomedical applications. Something not widely addressed in the state-of-the-art is the three-dimensional nanostructuring of the sensors to maximize exposure to active sites; this is indeed a promising pathway to make new photocatalytic materials reach higher performance levels [5,6,7]. SERS substrates could be fabricated using materials such as SiO_2_ [8], metal oxides [9], and noble metals. Among all of them, in most cases, the basis of plasmonic structures is gold (Au) or silver (Ag) [10,11,12,13,14,15]. Thanks to lower losses and a larger electron mean-free path, Ag offers a much higher field enhancement than gold [16]. On the other hand, Ag nanostructures suffer from low chemical stability—they easily sulfidize and oxidize under ambient conditions affecting their performance and limiting long-term stability, which is particularly important in sulfur-rich biological environments [17,18]. Previously, we showed that Ag nanostructures could be chemically stabilized by coating with a self-assembled monolayer (SAM) via sulfur bonds [19].

Importantly, the sensors, especially for bioapplications, should be mechanically stable and reusable, since it helps to minimize waste generation [20] and allows us to consider the prospect of implantable SERS platforms. Moreover, it is highly desirable that the substrates are flexible [15,21]. There are several widely used approaches to create flexible SERS substrates, including inkjet printing, sputtering, wet chemical synthesis, chemical immobilization, and so on [14,15,21,22]. Polymers, in this regard, are the most promising to be used as the base. Unfortunately, conventional methods to fabricate substrates fulfilling all these criteria, particularly for in vivo applications, do not result in robust structures. In this work, we aim to design a 3D structured, flexible sensor for photocatalytically driven SERS sensing. We based it on our previous findings on the laser-induced metal/polymer composite (LIMPc) formed by intermixing aluminum nanoparticles and polyethylene terephthalate (PET) [23]. We transformed our approach to integrate Ag nanoparticles into PET in a facile and single-step way by laser processing. Laser processing gives a benefit in terms of arbitrary shape patterning of robust plasmonic composite.

The complexity of the biological environment imposes the development of surface functionalization, providing selective analyte adsorption or serving as a molecular probe. Selecting the suitable probe in which spectra or spectral changes inform the sensor user about the chemical changes is a key issue in terms of selectivity and sensitivity in a complex environment.

The molecular probes are typically the thiol-based SAMs that participate in a chemical reaction with their environment, resulting in changes to their spectral signatures [24,25,26]. An exciting class of such probes is the one with the chemical changes triggered by the plasmonic structure itself via the photocatalytic mechanism: a novel and not fully understood process yet [27]. The mechanism behind plasmonic photocatalysis is a subject of intense debate on the role of charge transfer, photothermal effect, or their combination in the fundamentals [28,29,30]. Most photocatalytic reactions depend on the chemical environment, such as solvent [31,32]. One of the most studied plasmon-driven photocatalytic reactions is the reduction and dimerization of 4-nitrobenzenethiol (4-NBT) SAM, which can additionally be used as a molecular Raman probe and photocatalytic activity indicator. The photocatalytic reduction of 4-NBT to 4,4′-dimercaptoazobenzene (DMAB) is a complex reaction previously reviewed in the literature [33,34]. Driven by laser irradiation, the reaction depends on several parameters, including laser wavelength, exposure time, laser power, and environmental conditions such as pH. This sensitivity to pH was exploited for sensing intracellular and extracellular components, and reactive oxygen species, helping to detect different analytes in vivo [35] and in vitro [36].

Our Ag-based plasmonic composite has a high surface area and a large density of hot spots on top. To further serve as SERS photocatalytic sensor, it was functionalized with 4-NBT SAM as a Raman probe. In such architecture, there is no need for the analyte to be Raman active or to provide a strong signal. Instead, sensing occurs through the perturbation of the analyte molecules on the 4-NBT monolayer, driving plasmonic photocatalytic reactions [37]. Our flexible multifunctional material shows for the first time the possibility of filling the gap between plasmonic sensing and ultra-robust flexible, structural, and wearable electronics in an efficient, large-scale, and inexpensive way.

## 2. Results and Discussion

### 2.1. Development of a Novel Sensing Platform

In this work, we aim to develop a new photocatalytic sensing platform and design a robust flexible 3D plasmonic sensor aiming to fulfill the criteria for scalable, inexpensive, and free-form fabrication. Our concept is summarized in Figure 1 and is based on the sequential photothermal and chemical processing steps. We start by depositing a silver nanoparticle film on a PET substrate and its laser processing from the top. The optical images before and after the processing (Appendix A) do not reveal crucial changes in the morphology at the microscale; however, there are several important processes that are taking place. At this stage, Ag NPs film is critical to convert incoming photons into heat, which triggers changes in the underlying polymer substrate. One of the reasons we used PET is that it has a relatively low melting temperature, ca. 260 °C. The local temperatures due to photothermal heating exceed the PET melting point and initiate Ag NPs penetration into PET liquid phase. Until now, the same way as shown in our previous work using aluminum NPs, where laser integration of metal nanoparticles also initiated carbonization and formation of laser-induced graphene [23]. The crucial difference this work shows is that we are not reaching the PET pyrolysis temperatures, avoiding polymer carbonization. Such processing results in an electrically conductive network and simultaneous metal nanoparticle firm integration into the polymer surface. Moreover, we provide experimental confirmation for the physicochemical processes behind our material’s formation using high-speed video recorded during laser processing (Appendix A). This nanostructured silver layer generated at the top of the laser-induced metal–polymer composite (henceforth AgLIMPc) shows excellent electrical performance (0.16 ± 0.05 Ohm·sq^−1^). Such sheet resistance constitutes electrical conductivity four orders of magnitude lower than Al-based LIMPc [23]. In addition, mechanical robustness helps AgLIMPc to resist ultrasound cleaning, allowing not only reusability and disinfection with one of the harshest cleaning methods available but also the removal of nanoparticles not integrated into PET.

Thanks to precise control of temperatures reached during the laser treatment, we hypothesized that we could also minimize carbonization and leave the top Ag surface intact. However, previous laser-induced integration reports showed that carbonization and engulfing by the melted PET substrate were essential for a firm integration of the nanoparticles into the substrate. Here, we aim to resolve this dilemma: obtaining a pristine top silver layer without polymer capping or carbonization while achieving mechanical integration to the substrate. Our concept, illustrated in Figure 1a, is based on the spatially selective LIMPc formation at the Ag NPs/PET interface with nanoparticle sintering at the AgNP/air interface. These sintered Ag clusters serve as plasmonically active hotspots responsible for the electromagnetic field enhancement characteristic of such highly spatially confined locations [38]. The suggested mechanism of plasmonically active surface formation was confirmed using scanning electron microscopy (SEM) (Figure 1b), where the sintered Ag “forests” could be clearly observed. Low magnification SEM image (Appendix A) and energy-dispersive X-ray spectroscopy (EDX) maps (Appendix A) revealed the homogeneous distribution of forest-like structures with the dominant content of Ag (96.5%).

Silver is widely used for surface-enhanced Raman spectroscopy (SERS) sensors [39]. For our substrates, it is reasonable to expect that the integrated and sintered Ag “forests” would result in plasmonic enhancement. Therefore, the last fabrication step was forming a 4-NBT SAM on the three-dimensional Ag “forests” to be used as a photocatalytically active Raman probe. 4-NBT is an excellent molecule for this purpose as it undergoes a plasmon-induced photocatalytic conversion to 4-aminobenzenethiol (4-ABT) and 4,4′-dimercaptoazobenzene (DMAB), which could be simultaneously induced and tracked with Raman spectroscopy (Figure 1c).

### 2.2. Advancing toward Robust and Stable Sensors

For practical, real-life applications, we must make sure that our SERS sensor not only shows great performance but also is structurally and mechanically robust as well as reusable. To study this, a sample was sonicated for successive periods of 1, 2, 3, 5, and 10 min. Before and after each sonication step, the regions of the sensor were analyzed in 10 × 10 µm Raman maps to evidence changes induced by ultrasonication. The Raman maps normalized to the v_s_(NO_2_) vibrational mode (~1347 cm^−1^) in Figure 2a show the partial detachment of some Ag clusters after sonication (Appendix A). Even though we did not expect that AgLIMPc/4-NBT would remain functional after such harsh treatment, there was no critical degradation of the sensor signal within 10 min of sonication. The detachment of some clusters at different sonication times is reflected in a minor decrease in the absolute Raman signal intensity shown in Figure 2b. However, the decrease is not significant enough to compromise the signal acquisition with all spectra and intensities. The signal remains in the same order of magnitude without changes in signal/noise ratio since the remaining Ag clusters still have the high hotspot density responsible for signal enhancement. Although, the delamination of clusters can impact the uniformity of Raman signal intensities across the sensor surface, even though it does not significantly affect sensor performance.

The electrical conductivity and flexibility of our sensors are demonstrated by the results in Figure 2c. Deforming the sensor for over 1000 cycles did not result in the degradation of its electrical properties, which is shown by powering the LED through a flexible circuit (See Figure 2d). Moreover, our fabrication design is straightforward and offers spatially confined free-form patterning without the need for masks as in conventional lithography. Altogether, these advantages have strong implications for multifunctional sensor development [29,40,41].

Overall, compared to state-of-the-art, besides long-term stability and reusability, the base of our 3D-nanostructured plasmonic sensor is a polymer substrate, which makes it compatible with flexible electronics and large-scale production. 

We used our AgLIMPc/4-NBT sensing platform to investigate the possibility of mediating 4-NBT photocatalytic transformations and use this effect for probe-mediated SERS sensing of target analytes. This approach could be extremely sensitive since chemical transformations of 4-NBT to 4-ABT or DMAB and its intermediates are readily visible in the Raman spectra.

### 2.3. Mediation of Photocatalytic Reactions

The photocatalytic activity of AgLIMPc/4-NBT was studied under the influence of red laser light. However, to make a sensor reusable, it is necessary to control or avoid the irreversible photocatalytic conversion of 4-NBT to 4-ABT or DMAB. The facile way to minimize the molecule transformation is to adjust the laser power used in the Raman experiments. Indeed, DMAB (or 4-ABT) formation could be suppressed by keeping the laser power density below a threshold so that there is no dimerization or 4-ABT formation caused by other factors, and the effect we aim to observe is purely photocatalytic [42].

This lower power density was easy to achieve using a low magnification objective in the Raman microscope. We investigated the prostate cancer cells line PC3 liquid media extracted over different time periods (1–7 days) as a mediator of 4-NBT photocatalytic reactions, as sketched in Figure 3a. In such way, AgLIMPc/4-NBT could serve as a plasmonic sensor to tack changes in the surrounding environment caused by growth of this kind of cells. This cancer cell line has stable and rapid growth, reflected by noticeable changes in the culture medium. We show the sketch and an actual sample taken for analysis in Figure 3a. The Raman spectra of the sample with cell media were measured under ambient conditions. The spectra of the subsequent days up to day 6 and the ones from day 7 are shown in Appendix A and Figure 3b, respectively. To ensure that the observed effects originate from the cell’s metabolism products, we recorded control spectra from distilled water (further would be denoted as reference) and RPMI medium alone.

As expected, the 4-NBT signal dominates in every sample showing peaks at 1573 cm^−1^ for C-C stretching, 1347 cm^−1^ for symmetric NO_2_ stretching, 1109 cm^−1^ for C-N stretching, and 855 cm^−1^ for NO_2_ deformation (Figure 3b and Appendix A) [43,44]. The peak observed at 1078 cm^−1^ originates from C-S stretching, while the one at 722 cm^−1^ arises from Ag compounds. Both peaks also appear for Ag functionalized with 4-NBT and are typically observed in SERS spectra on Ag SERS substrates [44,45].

We recorded the Raman spectra from day 0 up to day 7 of cell cultivation. All the spectra were normalized to the C-C band, which is common to reactant and product molecules. Comparing the spectra from reference and cell media (CM) we did not observe any significant changes for days 1–6. However, on day 7 we see the appearance of three new bands at 1140 cm^−1^ (βC-H), 1390, and 1433 cm^−1^ (νN=N) coming from the 4-NBT dimerization product DMAB [46]. 4-NBT dimerization is a complex reaction with different possible pathways. These pathways could be triggered by environmental protonation or the presence of reactive oxygen species (ROS). In the first place, for every single pathway, the presence of H^+^ is important since the nitro group would capture it for further oxygen cleaving [36]; however, an acidic environment would not necessarily lead to the dimerization, 4-ABT (4-aminobenzenethiol) could be a reduction product, so to achieve dimerization via 4-ABT formation, the ROS species are needed [47].

In our experiment, the appearance of DMAB peaks could be attributed to the changes in the CM pH, which starts at 7.5 for the control sample (RPMI medium) and becomes more acidic up to day 7 (pH = 6.25). Thus, we performed a control experiment with the Britton–Robinson (BR) buffer at different pH to see if the changes observed in the Raman spectra were coming from pH perturbations (Appendix A and Figure 3c). From the Raman spectra recorded in different pH, we see that acidic pH (5.1) promotes DMAB formation (Figure 3c), in good agreement with the literature [36]. However, in the case of cell-medium analysis, the changes are more pronounced. The DMAB formation was quantified using the Raman intensity ratio I_1140_/I_1180_, where the peak at 1140 cm^−1^ is attributed to DMAB, and the one at 1180 cm^−1^ (C-H bending mode) is common for both reactant and product. The intensity ratio for CM media is higher than the one for BR, suggesting that the high DMAB yield at day 7 of cell cultivation could not be explained only by the medium acidifying.

It is known that on the last day of cultivation when cells start to die, there is a release of different compounds due to membrane damage. Those compounds include reactive oxygen species [48], which activate pathways for dimerization where the oxidative agent is needed. For instance, that could be the formation of 4-ABT with its sequential oxidation to DMAB [47].

Although quantitative analyses should be pursued, these results already evidence the success of our concept with AgLIMPc/4-NBT as an indirect molecular SERS sensor compatible with flexible electronics. These flexible plasmonic sensors allow tracking changes in the cancer cell media that otherwise are not visible with Raman or with conventional SERS. Moreover, we observed drastic spectral changes only when cells started to die, meaning that the fabricated sensor could be useful for cancer treatment monitoring rather than for analysis of its growth. This motivates us to adapt our sensor design and materials for future works on SERS detection in complex human body fluids and expand our technology towards other types of cells, which, thanks to the flexibility, low cost, and lightweight nature, could make a significant impact on healthcare. We also consider exploring other approaches for quantification, such as combining SERS with electrochemistry [25] or merging with slab waveguide technology [49].

## 3. Materials and Methods

AgLIMPc formation. As a base, commercial PET sheets with 0.7 mm thickness were used as received. PET chemical composition was proved by Raman spectroscopy (Appendix A). Thin films of Ag nanoparticles on PET were drop-casted from AgNPs ethanolic suspension (60 mg mL^−1^) with an areal density of 0.71 μL mm^−2^ and dried under ambient conditions. The suspension was sonicated for 20 min before the deposition. Laser patterning to obtain LIMPc was performed using a computer-controlled blue laser.

4-NBT functionalization: 4-NBT (1 mM) solution was prepared in a 1:1 water/ethanol mixture. AgLIMPc samples were immersed in a 4-NBT solution overnight to ensure SAM formation. On the next day, the samples were rinsed with ethanol and distilled water to remove physically adsorbed molecules and dried in ambient conditions [50].

Ultrasonication was performed using a 120 W ultrasound bath with a frequency of 40 kHz.

Scanning electron microscopy (SEM) with energy-dispersive X-ray spectroscopy (EDX) was performed using Quanta 200 3D, FEI with a step size of 10 eV.

4-point probe measurements: All the sheet resistance values were obtained by 4-point probe measurements. We used Potentiostat/Galvanostat P-45X with FRA-24M impedance modulus (Electrochemical instruments, Russia) in galvanostatic mode. Probes were set at a square with a 400 μm side, the current of 1 mA was applied along one side of a square, and potential was measured along the parallel side. Sheet resistance values were calculated using the following equation:(1)$$R=\frac{2\pi }{ln2}\cdot \frac{V}{I}$$
where *R* is sheet resistance, *V* is measured voltage drop, and *I* is applied current. The value presented in the text is the averaged value from three samples with the standard error.

Cell media. The prostate cancer cell line PC3 was selected to study the influence of the intracellular medium of cancer cells on the AgLIMPc/4-NBT substrate. Cells were cultivated in an RPMI 1640 medium (with glutamine, antibiotics penicillin-streptomycin) with fetal bovine serum at 37 °C in a CO_2_ incubator (5%). The medium was not replaced during the period of the experiment (7 days). Every day, an aliquot of ~1 mL of the media was collected, centrifuged, and placed in sterile containers. Then, the samples were kept in the fridge at +5 °C for the endpoint.

Raman measurements were performed using confocal Raman microscopy (NTEGRA SPECTRA). 633 nm laser was used for sample characterization using a 20x objective while exposure time was equal to 10 s. The laser power was 175 μW. For each sample reference spectrum was recorded in distilled water. From each sample, the signal was recorded from 5 spots and then averaged using OriginPro software.

Reusability: The samples were sonicated (120 W output power and 40 kHz) for 1, 2, 5, and 10 min. The 10 × 10 µm^2^ Raman map was recorded as a reference. Subsequently, Raman maps were recorded after each period of sonication.

Bending sensor performance was studied with a three-point flexural test. A 70 × 10 mm^2^ engraved stripe-shaped sample was used for this test. The distance between the two side points was adjusted to 20 mm. The middle point movement was controlled by a micro-step motor, and the resistance values were measured by Potentiostat-galvanostat P45-X. A constant current was applied to the sample, and a voltage drop during bending cycles was measured.

Laser-speckle visualization. A high-speed imaging system with a CuBr vapor brightness amplifier and a time resolution of 500 and 900 frames per second was used for this experiment. The images were made with the Helios-44M objective lens (F = 58 mm). The temporal resolution of the imaging system was 5 μm.

## 4. Conclusions

In this work we demonstrated for the first time the integration of plasmonically active Ag clusters into a flexible polymer by single-step laser patterning. The laser sintering of silver nanoparticles leads to the formation of plasmonic Ag “forests”. We further performed 4-NBT monolayer functionalization that allows electromagnetic field enhancement essential for SERS, which helps to evidence the plasmon-induced catalytic activity by tracking the transformation of 4-NBT to DMAB. 4-NBT was also further used as a molecular probe in an integrated, flexible device sensitive to cancer cell death. This sensor platform showed remarkable resilience against harsh treatments such as ultrasonication and thousands of bending cycles. This innovative device assembly contributes to the development of wearable sensors that can identify complex chemical substances by the influence of an analyte on the molecular Raman probe. Moreover, artificial intelligence algorithms will boost this strategy to assist in real-time tracking, recognize spectral patterns, and link them to specific substances in the human body and their concentrations [51]. This is one of the fascinating challenges we aim to tackle next.

## Figures and Tables

**Figure 1 nanomaterials-13-00923-f001:**
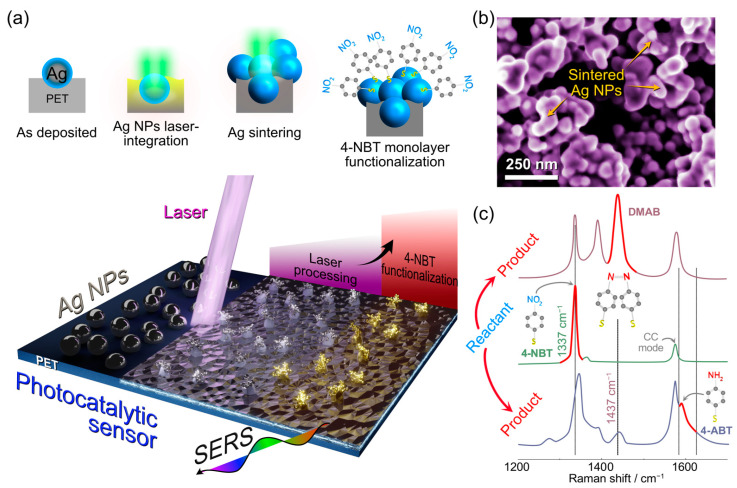
(**a**) Schematic concept of the 3D molecular plasmon-active structures. A thin film of Ag NPs is integrated into the polymer substrate by laser processing. The resulting three-dimensional Ag microforests are then functionalized by self-assembled monolayers of 4-NBT. (**b**) SEM top-view image of AgLIMPc showing the surface of the silver/PET composite. The inset SEM image was obtained from the top of a sample with closely spaced Ag nanostructures into clusters forming plasmon-active hotspots. (**c**) Characteristic spectra of the 4-NBT Raman probe and the main vibration modes of the different species (4-ABT and DMAB) can evolve during SERS. All spectra in (**c**) were drawn from actual experimental results.

**Figure 2 nanomaterials-13-00923-f002:**
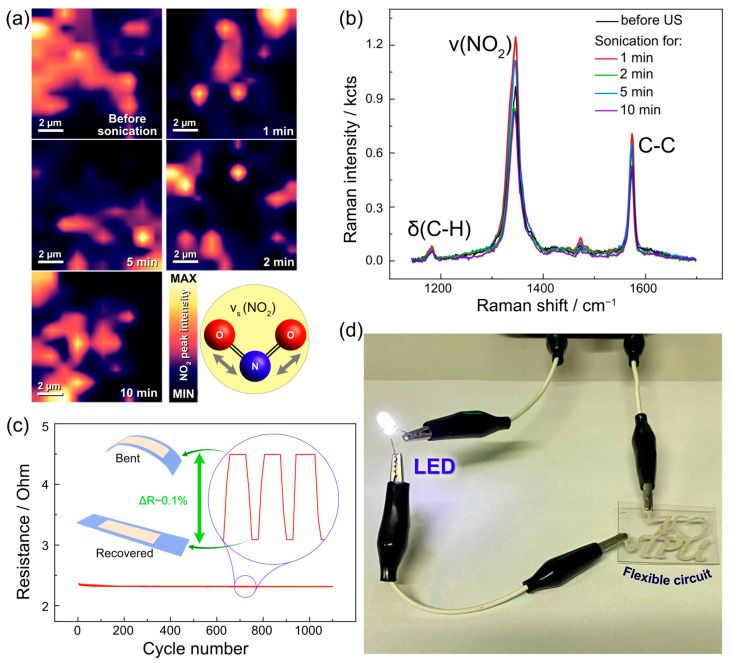
(**a**) The study of reusability and robustness by placing the substrate into the ultrasonic bath for different time periods: 0, 1, 2, 5, and 10 min. The performance was estimated with a Raman map of NO_2_ vibrational mode. (**b**) Raman spectra of the sample after each ultrasound session in absolute scale. (**c**) Bending test of the AgLIMPc. (**d**) Demonstration of the ability to create a flexible circuit by powering LED through AgLIMPc sample.

**Figure 3 nanomaterials-13-00923-f003:**
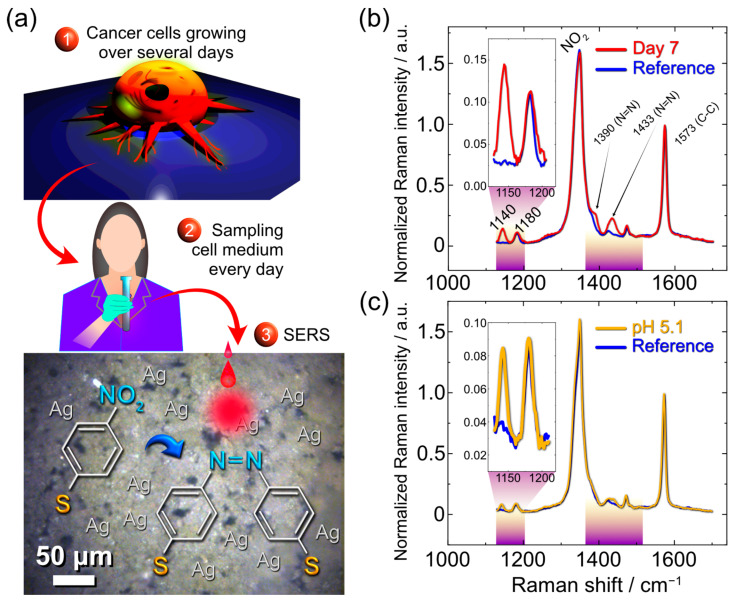
(**a**) Schematics of the experiments with prostate cancer PC3 cell line RPMI media. Culture media were extracted every day. (**b**) Raman spectrum of cell media extracted on the seventh day on AgLIMPc/4-NBT substrate with the reference spectrum of water on the same substrate. The regions of interest with the main changes are marked in the spectra. (**c**) Spectra comparison of the AgLIMPc/4-NBT covered with a pH buffer (pH = 5.1) and water.

## Data Availability

Data available from corresponding author by request.

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
