# Peer review of "Molecular Plasmonic Silver Forests for the Photocatalytic-Driven Sensing Platforms"

_nanomaterials, 2023, doi:10.3390/nano13050923_

Round 1

Reviewer 1 Report

The authors have developed 3D plasmonic nanostructures/polymer surfaces with 4-NBT as a molecular probe for SERS detection. The authors have investigated the sensor’s performance to identify the cell death of prostate cancer. The laser-driven nanoparticles/polymer intermixing has revealed free-form electrically conductive composites. Overall, this work can inspire more material design ideas of plasmonic-based flexible electronics. Therefore, I would like to recommend this work to publish in Nanomaterials. Below are some comments for the authors.

1. For Figure 2b of Raman spectra, the characteristic peaks of Raman peaks should be labeled.

2. For Figure S2, the elemental percentages of Ag and C should be provided to reveal their composition.

3. For the introduction “The basis of a plasmonic structure is typically gold (Au) or silver (Ag)”, more references could be cited to broaden the introduction.

https://doi.org/10.1016/j.jhazmat.2020.124617

Author Response

The point-by-point response is in the pdf here attached

Reviewer 2 Report

The work proposed by Fatkullin et al. on plasmonic silver nanoparticles embedded in a flexible substrate sounds very interesting towards low-cost and disposable devices. I have some minor comments:

- The introduction should better focus on the state of art of plasmonic/sers probes based on flexible substrates (like PET). For instance, easy to fabricate and low-cost devices can be developed through inkjet/3D-printing combined with metal (gold or silver) nanoparticles.

- What are the specifications of the used PET substrate? Have the authors considered the idea to use it a slab waveguide to excite plasmonic phenomenon? About this, the authours could refer to this work 10.1109/TIM.2018.2879170?

- The authors should improve some of the Section titles. Titles like "Towards the impressive sensor stability" are not accetable in my opinion.

- I suggest the authors to consider to enrich the results section, for instance by moving some of the figures from the supplementary to the main manuscript.

Author Response

The point-by-point response is here attached as a pdf file.

Reviewer 3 Report

The authors reported results on the integration of plasmonic Ag nanoparticles into a flexible polymer matrix by using a laser patterning technique. The authors show that the sintering of Ag nanoparticles leads to the formation of a plasmonic Ag Forests structure, with highly sensitive photocatalytic properties required for active SERS sensors. The authors evidenced the plasmonic properties of Ag Forests structures by monitoring the transformation of 4-NBT molecular probe to DMAB. Presented results show that the prepared sensor platform keeps the same performance even in harsh working conditions such as ultrasonication and after 1000 bending cycles, could have an impact on the monitoring of the cancer treatment process.

The authors present results that may be of interest to the readers of the journal, but a number of clarifications should be made to the following points and comments

- In line 113, the authors should describe the terms in the equation.

- In figure 1, the authors should provide the image of the nanoparticles just after deposition, to illustrate the evolution of the morphology after sintering. 

- In line 242, the authors mention Figure 4a, perhaps they mean Figure 3a.

- Figure 1c the spectra of the 4ABT compounds show similar peaks to those observed with DMAB, does this mean that there is a mixture of the two compounds? If this is true, I don't understand this statement in the manuscript text

"The easy way to minimize the transformation of the molecule is to adjust the power of the laser used in the Raman experiments. Indeed, the formation of DMAB could be suppressed by keeping the laser power density below a threshold [32]. This lower power density was easily achieved by using a low magnification objective in the Raman microscope "

- From Figure 2a, it appears that the number of detached nanoparticles is significant, can the authors comment on the minor decrease in absolute Raman signal intensity?

"The detachment of some clusters at different sonication times is reflected in a minor decrease in the absolute Raman signal intensity shown in Figure 2b."

Author Response

(The authors gave the same response as above.)

Round 2

Reviewer 3 Report

Dear authors,

The authors have responded to my comments and questions in a satisfactory manner, and therefore I recommend the publication of the revised version of the article.

Best regards